# Parental Psychological Flexibility and Children’s Behavior Problems in Rural Areas in Northeast China: The Mediation of Children’s Emotion Regulation

**DOI:** 10.3390/ijerph192315788

**Published:** 2022-11-27

**Authors:** Xiaoling Ren, Xiaoying Ren, Zhonglian Yan, Songhan Lu, Xiaohan Zhou

**Affiliations:** 1School of Education, Beihua University, Jilin 132013, China; 2Jinxue Primary School, Yanji 133099, China; 3School of Education, Harbin Normal University, Harbin 150025, China; 4Faculty of Education, Northeast Normal University, Changchun 130024, China

**Keywords:** parental psychological flexibility, children’s emotion regulation, children’s behavior problems, rural preschoolers

## Abstract

Children’s behavior problems are not conducive to their sustainable development. Therefore, it is of great value to explore the mechanism of relevant influencing factors on the behavior problems of rural preschoolers. This study aimed to reveal the direct effect of parental psychological flexibility on children’s behavior problems and the mediating effect of children’s emotion regulation. Based on simple random sampling, 355 caregivers (male = 31.25 years, SD = 9.78; 74.08% females; 9.01% bachelor degree) were recruited from eight rural kindergartens in three provinces in northeast China. With questionnaires, caregivers reported their parental psychological flexibility and assessed their children’s emotion regulation and behavior problems. SPSS 25.0 software was used for statistical data analysis. The results support our hypotheses, suggesting that parental psychological flexibility, emotional stability, and emotional regulation negatively predicted children’s externalizing and internalizing behavior problems. Meanwhile, emotional stability and regulation partially mediated the relationship between parental psychological flexibility and children’s externalizing and internalizing behavior problems. These findings provide a new perspective for preventing and intervening in preschoolers’ behavior problems.

## 1. Introduction

Children’s behavior problems (CBP) are abnormal behaviors that hinder their adaptation to society [1]. Perou et al. (2013) pointed out that 13–20% of children have psychological problems, and most of these psychological problems directly lead to CBP. These data showed an increasing trend compared with the early years [2]. Moreover, according to the Chinese Youth Development Report, about 30 million Chinese people under 17 have behavior problems [3]. Studies have shown that most individual behavior problems start early in childhood [4]. Behavior problems directly negatively affect children’s learning, and the behavior problems in early childhood will continue to adolescence or even adulthood if there is no intervention [5]. Robins and Price (1991) found that more than 70% of adults with antisocial behavior problems had strong antisocial behavior tendencies in childhood [6]. Studies have found that children from low-income families have worse behavior problems than those from high-income families [7]. Most rural families’ economy is fragile, so it is essential to explore the influencing factors of rural preschoolers’ behavior problems.

Previous studies mainly discussed factors affecting CBP are analyzed from the perspective of the developmental assets framework. Development assets refer to relevant experiences, relationships, skills, and values that can effectively promote individuals to achieve healthy development outcomes, which mainly include external assets and internal assets [8]. Ecological systems theory proffered that family is an essential external asset for individual development, and the interaction among family members is crucial to preschoolers’ development [9]. High levels of parental love and support [10], active communication, and parenting practices [11] were beneficial in reducing the incidence of children’s problem behaviors. Therefore, previous studies hoped to reduce CBP by improving parenting concepts, strategies, and methods. However, in recent years, the probability of individual behavioral problems is still increasing and showing a trend of younger age [12]. Moreover, studies have pointed out that the stability of parenting practices raised the difficulty of practical operation. It is not easy to fundamentally and continuously improve the parent–child interaction process and relationship quality by teaching parents some positive parenting concepts, strategies, and methods through simple training [11]. Parental psychological flexibility (PPF) is the basis of positive parenting practice [13]. The study of acceptance and commitment therapy in cognitive–behavioral therapy found that with the improvement of PPF, the attitude and behavior of parents are more positive, and the children also have a positive tendency [14]. These provide a new perspective for improving CBP and family education contexts.

Individual adaptability is one of the critical internal assets for personal development [9]. In recent years, the influence of individual regulation and attachment on CBP has been studied by researchers. Studies have found that preschoolers sometimes experience both externalizing and internalizing behavior problems such as aggression, depression, and social withdrawal. The low level of emotion regulation is the main characteristic of these children [15]. Children’s emotion regulation (CER) was positively correlated with their socialization [16]. Moreover, positive attachment can effectively promote the healthy development of children’s sociality and personality [17]. Avoidant and chaotic attachment had stronger predictions of externalizing behavior problems [18]. Therefore, the healthy development of children needs to help children form positive emotion regulation and attachment. Furthermore, previous studies found that CER mediated the association between attachment and depressive disorders [19]. In particular, the theory of cognitive development points out that preschoolers also have subjective initiative. In sum, whether the harmful external factors will affect preschoolers’ behavior problems needs their active self-regulation and control [20]. Based on the theory of cognitive development and ecological systems theory, this study intends to determine how PPF and CER affect their behavior problems. This study may provide new empirical support for the intervention of preschoolers’ behavior problems. In addition, it could improve family education and promote educational equity in China.

### 1.1. Parental Psychological Flexibility and Children’s Behavior Problems

Parental psychological flexibility (PPF) means that parents accept their negative thoughts, emotions, and impulses toward their children and insist on effective parenting methods [13]. PPF covers cognitive defusion, committed action, and acceptance. Cognitive defusion refers to parents’ ability to consciously separate their negative emotions and thoughts from their parenting behavior so that their feelings or thoughts do not control the parenting behavior. Committed action means that parents can fully respect their children’s wishes and allow them to maintain their independence appropriately. These may increase the parents’ worry and anxiety for their children, but parents can effectively regulate themselves. Acceptance means that parents can accept the emotions and thoughts that are painful to them during parenting. They can recognize and allow these as part of parenting [13]. Externalizing and internalizing behavior problems are two primary domains of CBP [1]. Children’s externalizing behavior problems are easily observed behaviors that do not conform to social norms, usually manifested as hurtful or destructive behaviors. Children’s internalizing behavior problems refer to their unhealthy mental state, primarily displayed as extremely negative emotions or moods. Moreover, children’s externalizing and internalizing behavior problems include many specific contents. Based on preschoolers’ characteristics and the availability of survey tools, children’s externalizing behavior problems in this study mainly have disciplinary violations and aggressive behaviors. Children’s internalizing behavior problems mainly include social withdrawal, depression, and anxiety.

The family is the central place for children’s activities, and the family education contexts directly affect children’s physical and mental health development [21]. For instance, warm parenting reduces the risk of CBP [22]. The lower the interaction process and relationship quality between parents and children, the more likely children are to have psychological and physical problems [23]. Maternal dysphoric mood predicts behavioral difficulties in preschoolers [24]. PPF help shapes positive parenting practices and family education contexts [25]. In China, the behavior problems of rural children were more serious, and their parents fell behind urban parents in terms of parenting philosophies and strategies [26]. PPF may affect CBP, especially rural children, by regulating parenting pressure, behavior, and attitude. In particular, some empirical studies have analyzed the influence of PPF on CBP. PPF was negatively associated with chronic pain in children [27]. PPF correlated significantly with functional disability and depression in adolescents [28]. Individual chronic pain, functional disability, and depression are all internalizing behavior problems. Furthermore, PPF was significantly related to youth externalizing behavior problems [25]. As mentioned above, PPF plays a crucial role in parenting practice and may affect children’s externalizing and internalizing behavior problems.

### 1.2. Children’s Emotion Regulation and Children’s Behavior Problems

Children’s emotion regulation (CER) is how they manage their emotional states [29]. CER includes emotional stability and emotional regulation. Emotional stability mainly refers to effectively controlling mood swings and negative emotions. Emotional regulation specifically includes children’s emotional understanding and empathy [29]. The development of emotion regulation is an essential link in children’s socialization. Successful emotion regulation is conducive to children’s positive social adaptation [30]. Predicting CER suffering on their behavior problems has received more attention in empirical research. The premise of CBP was emotional regulation difficulties [31]. CBP can be influenced by shaping their emotion regulation [32]. In addition, some researchers have analyzed the effect of CER on children’s externalizing and internalizing behavior problems. CER was a key factor affecting their destructive behaviors [33]. Children’s destructive behavior belongs to externalizing behavior problems. Nigg (2006) noted that children’s depression was closely related to their lack of emotional regulation strategies [34]. Preschoolers with anxiety disorders reported more emotional regulation disorders than their peers [29]. Children’s depression and anxiety disorders belong to internalizing behavior problems. Moreover, children’s emotional regulation affects their sociality and predicts their externalizing and internalizing behavior problems [35]. However, few studies have analyzed the relationship between the two dimensions (emotional regulation and stability) of CER and their behavior problems. Emotionally unstable children were often rejected by others [16]. Children’s empathy was positively correlated with the development of their social skills [36]. Children with difficulty understanding emotions and emotional instability often fail to meet the demands of their social environment [37], which intensifies the risk of externalizing and internalizing behavior problems.

### 1.3. The Mediating Effect of Children’s Emotion Regulation

Parents are important nurturers, socializers, and role models of preschoolers’ learning and development [9]. Moreover, rudimentary behavioral problems and emotion regulation skills are evident early in life [2]. Although internal factors such as genetics, attachment, temperament, and neurobiology seem to contribute to CER, scholars have agreed that CER is also influenced by family education contexts [38]. Parke (1994) suggested that parents’ emotions and performance implicitly teach individuals which emotions are accepted [39]. As such, parents’ positive emotions improved CER, and parents’ negative emotions negatively affected CER [40]. Parent–child interaction positively impacts CER [41]. Furthermore, Spruijt et al. (2018) pointed out that excellent parenting strategies were essential to satisfy the needs of children, which might positively affect CER [42]. Daks and Rogge (2021) pointed out that psychological flexibility directly affected parenting interaction and emotion [43]. In sum, PPF may shape children’s emotional regulation by influencing parenting practices and attitudes.

Given that PPF and CER may be interrelated, CER is related to their behavior problems. CER may be a conduit from PPF to CBP. We are unaware of studies examining CER mediating the associations between PPF and CBP. However, several studies on the role of parenting practices and attitudes in CBP have examined similar indirect pathways. Findings underscored that supportive parenting shifted children from antisocial trajectories by shaping their physiological regulation and behavioral adjustment [22]. Eisenberg et al. (2001) found that the relationship between maternal emotion expression and children’s externalizing behavior problems was mediated by CER [44]. Emotion regulation of children aged 3–7 accounted for the indirect association between maternal emotional awareness and children’s internalizing and externalizing problems [45]. CER mediates the associations between parental emotion socialization and children’s psychopathological symptoms [46]. These studies have elucidated the mediating role of CER between parenting practices and attitudes and CBP. Psychological flexibility is the cornerstone of parenting emotion, exercise, and concepts [13]. The role of CER as a mediator in the relationship between PPF and CBP is still a meaningful direction.

### 1.4. Present Research

This study will present the mechanism of PPF, CER, and CBP for rural preschoolers. It can provide new empirical support to ensure children’s healthy growth, improve family education contexts, and promote educational equity. Based on study questions and previous studies, we hypothesized that (1) Hypothesis 1 PPF, emotional stability, and emotional regulation were negatively associated with children’s externalizing and internalizing behavior problems. Furthermore. Hypothesis 2 Emotional stability and emotional regulation mediate between PPF and children’s externalizing behavior problems. Hypothesis 3 Emotional stability and emotional regulation mediate the associations between PPF and children’s internalizing behavior problems. The hypothesis model of this study is shown in Figure 1.

## 2. Materials and Methods

### 2.1. Participants

This study invited eight kindergartens in three provinces in northeast China, including Jilin, Haerbin, and Liaoning, by simple random sampling. We advertised this survey to the teachers working in these kindergartens and requested the kindergarten teachers invite caregivers to participate in this survey. The caregivers reported the variables required to be measured using a questionnaire. In addition, we randomly placed three items throughout the questionnaire to reduce social desirability. Individuals who answered an average of two of these three items in a contradictory fashion were excluded from the sample. After the caregivers agreed to participate in this study, the kindergarten teachers distributed questionnaires when caregivers picked up their children from kindergarten and guided the caregivers to fill in the questionnaires. Three hundred ninety-five questionnaires were distributed to the caregivers, and 355 valid questionnaires were recovered, with an effective rate of 89.87%.

Among them, 62.53% of the questionnaires were completed by mothers, 25.36% by fathers, and 12.11% by grandmothers. A total of 244 of these caregivers worked in agriculture. In addition, 195 caregivers reported a middle school diploma or lower, 128 had a high school diploma, and 32 had a college degree. The average age of the preschoolers in this study was 59.97 months, and 192 were female.

### 2.2. Measures

#### 2.2.1. Parental Psychological Flexibility Questionnaire

Li et al. (2018) translated and revised the parental psychological flexibility questionnaire [47], which was compiled by Burke et al. (2015) [13]. The translated and edited parental psychological flexibility questionnaire can effectively assess the level of PPF of Chinese caregivers [47]. The parental psychological flexibility questionnaire was commonly used to evaluate the PPF of pre-adolescents and adolescents [15]. However, Wu et al. (2018), Fu et al. (2018), and Wang et al. (2021) all confirmed that the parental psychological flexibility questionnaire has good reliability and validity in assessing the psychological flexibility among parents of preschoolers [11,48,49]. Therefore, this study used a 16-item parental psychological flexibility questionnaire translated and modified by Li et al [47]. to assess psychological flexibility among caregivers of preschoolers. All items in the scale were rated on a 5-point Likert scale (from 1 = ‘strongly disagree’ to 5 = ‘strongly agree’). The Cronbach’s alpha of the total sample was 0.98. The fewer scores, the lower PPF the caregivers had.

#### 2.2.2. Child Behavior Checklist

The caregivers assessed children’s externalizing and internalizing behavior problems using the 55-item child behavior checklist. Zhang et al. (2008) translated and revised the child behavior checklist [50], which was compiled by Achenbach et al. (1983) [1]. The translated and edited child behavior checklist has shown reliability for Chinese caregivers [50]. There were 33 items in the children’s externalizing behavior problems subscale, and Cronbach’s alpha of this subscale was 0.97. The children’s internalizing behavior problems subscale consisted of 22 items, and Cronbach’s alpha of this subscale was 0.98. All items in the scale were rated on a 0–2 scale (from 0 = ‘strongly inconsistent’ to 2 = ‘strongly consistent’). The scores were positively correlated with the levels of children’s externalizing and internalizing behavior problems. The Cronbach’s alpha of the total sample was 0.98.

#### 2.2.3. Emotion Regulation Checklist

Zhu et al. (2020) translated and revised the emotion regulation checklist [51], which was compiled by Shields and Cicchetti (1997) [29]. The translated and edited emotion regulation checklist can effectively assess the level of CER by Chinese caregivers [51]. The emotion regulation checklist, translated and revised by Zhu et al. (2020) [51], contains 24 items. There were two scales in the emotion regulation checklist. The emotional stability subscale comprised 16 items, and Cronbach’s alpha of this subscale was 0.94. There were 33 items in the emotional regulation subscale, and Cronbach’s alpha of this subscale was 0.83. All items were rated on a 4-point scale (from 1 = ‘Rarely/Never’ to 4 = ‘Almost Always’). Moreover, higher scores reflected greater levels of emotional stability and emotional regulation. The Cronbach’s alpha of the total sample was 0.94.

### 2.3. Procedure

First, we contacted the sampled kindergartens through the local education department and explained the purpose and content of this study to obtain their permission and cooperation. Second, through kindergarten teachers, we invited some rural caregivers to complete the questionnaire. Before participating in the survey, caregivers received a letter that provided information on the purpose and content of this survey. The letter also stressed the anonymity and confidentiality of the information collected. The questionnaires were issued to caregivers after they decided to participate in the survey. Third, caregivers were asked to independently answer the questionnaire according to their and their children’s situation. Then, we processed the collected data and filtered out invalid data. Moreover, this study was approved by the Ethics Committee of the first author’s university.

### 2.4. Date Analysis

SPSS 25.0 software for the statistical data analysis in this study. Pearson correlation was used to analyze the correlation among variables. Multiple regression analysis was used to analyze the effects of PPF and CER on their behavior problems. The multiple regression analysis used PPF as the independent variable. The dependent variables were children’s externalizing and internalizing behavior problems. Emotional regulation and stability were the mediating variables. Furthermore, mediation analyses contained the following covariates: caregiver gender, caregiver category, caregiver education, caregiver job, child age, and child gender. Suppose the influence of the independent variable on the dependent variable is transmitted through the third variable. In that case, the third variable is the mediator between the independent and dependent variables [52]. Hayes (2013) clarified the procedure and methodology of mediation analysis [53]. To examine whether emotional regulation and stability were potential mediators of the relationship between PPF and children’s externalizing and internalizing behavior problems, we followed the method outlined by Hayes (2013). The level of significance was set at *p* < 0.05.

## 3. Results

### 3.1. Correlation Analysis of Parental Psychological Flexibility, Children’s Emotion Regulation, and Children’s Behavior Problems

Table 1 shows that PPF was positively associated with emotional regulation (r = 0.50, *p* < 0.01) and emotional stability (r = 0.49, *p* < 0.01) and negatively associated with children’s externalizing behavior problems (r = −0.41, *p* < 0.01) and children’s internalizing behavior problems (r = −0.54, *p* < 0.01). Emotional regulation was negative associations with children’s externalizing behavior problems (r = −0.41, *p* < 0.01) and children’s internalizing behavior problems (r = −0.54, *p* < 0.01). Emotional stability was negative associations with children’s externalizing behavior problems (r = −0.78, *p* < 0.01) and children’s internalizing behavior problems (r = −0.81, *p* < 0.01).

### 3.2. Main Effects of Parental Psychological Flexibility and Children’s Emotion Regulation on Children’s Behavior Problems

Table 2 shows that the adjusted R^2^ value of the regression model is 0.65, indicating that the predictive power of PPF, emotional regulation, and emotional stability on children’s externalizing behavior problems reaches 65%. Moreover, PPF is a significant negative predictor of children’s externalizing behavior problems (β  = −0.08, *p*  <  0.01). Emotional regulation (β = −0.20, *p*  <  0.01) and emotional stability (β  = −0.79, *p*  <  0.01) negatively predicted children’s externalizing behavior problems.

Table 3 shows that the adjusted R^2^ value of the regression model is 0.72, indicating that the predictive power of PPF, emotional regulation, and emotional stability on children’s internalizing behavior problems reaches 72%. Moreover, as shown in Table 3, PPF is a significant negative predictor of children’s internalizing behavior problems (β  =  −0.13, *p*  <  0.01). Emotional regulation (β = −0.47, *p*  <  0.01) and emotional stability (β  =  −0.68, *p*  <  0.01) negatively predicted children’s internalizing behavior problems. Hypothesis 1 was supported.

### 3.3. The Mediating Roles of Emotional Regulation and Stability

Table 4 and Figure 2 confirm that emotional regulation and stability mediated the relationship between PPF and children’s externalizing behavior problems. The results illustrate that under the influence of mediation, PPF has a direct (95% CI [−0.15, −0.01]) significant impact on children’s externalizing behavior problems. The directing effect value was −0.08. Moreover, emotional regulation significantly mediated the relationship between PPF and children’s externalizing behavior problems (95% CI [−0.12, −0.03]). The mediating effect value was −0.06, accounting for 14.63% of the total mediating effect. Furthermore, emotional stability significantly mediated the relationship between PPF and children’s externalizing behavior problems (95% CI [−0.40, −0.24]). The mediating effect value was −0.27, accounting for 65.85% of the total mediating effect. Hypothesis 2 was supported.

Table 5 and Figure 3 confirm that emotional regulation and stability mediated the relationship between PPF and children’s internalizing behavior problems. The results illustrate that under the influence of mediation, PPF has a direct (95% CI [−0.18, −0.06]) significant impact on children’s internalizing behavior problems. The directing effect value was −0.08. Moreover, emotional regulation significantly mediated the relationship between PPF and children’s internalizing behavior problems (95% CI [−0.20, −0.11]). The mediating effect value was −0.15, accounting for 30.00% of the total mediating effect. Furthermore, emotional stability significantly mediated the relationship between PPF and children’s internalizing behavior problems (95% CI [−0.31, −0.18]). The mediating effect value was −0.23, accounting for 46.00% of the total mediating effect. Hypothesis 3 was supported.

## 4. Discussion

This study examined the relationship among PPF, two dimensions (emotional regulation and stability) of CER, and two dimensions (children’s externalizing and internalizing behavior problems) of CBP. In addition, we examined the mediating roles of emotional regulation and stability on the relationship between PPF and children’s externalizing and internalizing behavior problems. Our hypotheses were supported.

### 4.1. Parental Psychological Flexibility and Children’s Emotion Regulation Significantly Negatively Predicted Children’s Behavior Problems

Our results showed that PPF significantly and negatively affected children’s externalizing and internalizing behavior problems. Some studies also focused on the relationship between PPF and CBP. For example, Brassell et al. (2016) found that the higher the PPF, the less the children’s externalizing and internalizing behavior problems [25]. In particular, PPF can regulate the degree of child distress [54]. Compared with previous studies, this study pays more attention to the effect of PPF on preschoolers’ development. Preschoolers’ psychology and physiology are constantly developing and changing, and they are more likely to form externalizing and internalizing behavior problems in the face of pressure and conflict [55]. Higher PPF could reduce the parenting pressure and let parents adopt effective parenting strategies, thus significantly reducing the risk of undesirable stimuli in the family education contexts [25]. Therefore, higher parenting flexibility can create positive family education contexts and thus prevent the development of preschoolers’ behavior problems. On the contrary, lower psychological flexibility leads to stronger parental control over children, and strict controls could pressure children greatly. Greater pressure and control from the family will increase the probability of preschoolers’ behavior problems [56]. Furthermore, the findings of this study also support the ecosystem theory that preschoolers’ development is influenced by factors from their families [57]. In the future, we can improve preschoolers’ externalizing and internalizing behavior problems by intervening in PPF.

This study is consistent with previous studies that found significant adverse effects of CER on their externalizing and internalizing behavior problems. For example, empirical studies pointed out that CER difficulties caused their externalizing behavior problems [34,35]. Moreover, negative emotional experiences restrict CER and cause internalizing behavior problems [32]. In particular, Greenberg et al. (1995) constructed the ‘Promoting Alternative Thinking Strategies’ curriculum to promote preschoolers’ emotion regulation [58]. After receiving this course, preschoolers’ emotion regulation improved significantly, reducing their behavior problems. However, unlike previous studies, this study focused on the effects of two dimensions (emotional regulation and stability) of CER on children’s externalizing and internalizing behavior problems. The weaker children’s emotional regulation, the more likely they are to produce anger and anxiety in the face of negative stimuli [59]. Moreover, the more children’s emotionally stable, the weaker the children’s response to external stimuli [60]. Therefore, the degree of children’s emotional stability and regulation ability determines the suitability of children’s social behavior. In addition, the findings of this study also support the theory of cognitive development that individuals can control their behavior and psychology through self-regulation [7]. Furthermore, this study pays more attention to rural preschoolers. Rural preschoolers live in a more complex environment and are easily exposed to harmful stimuli, so we should focus on improving rural preschoolers’ emotion regulation.

### 4.2. The Mediating Effect of Emotional Regulation and Stability

Our results showed that emotional regulation and stability partially mediate the relationship between PPF and children’s externalizing and internalizing behavior problems. That is, PPF can partially reduce children’s externalizing and internalizing behavior problems by enhancing their emotion regulation. Few previous studies have explored the relationship between PPF, CER, and their behavior problems. The family was central to children’s social learning and an essential factor affecting children’s development [61]. Parenting psychological flexibility positively influences parents’ understanding, attitude, and strategies toward their children, which contributes to suitable family education contexts [13]. Moreover, children’s development is a process of individual active construction, which is realized in the interaction between the individual and the environment [20]. Parents with low psychological flexibility are more inclined to raise their children through violence [13]. However, these unfit parenting practices and attitudes further increase the risk of children’s externalizing and internalizing behavior problems by depleting their emotion regulation [62].

Furthermore, empirical studies have shown the relationship between attachment, emotion regulation, and depression. CER was a mediator between attachment to their parents and depression [63]. The regulation of sadness did mediate the relationship between attachment and depressive symptoms [64]. Attachment theory proposes that different attachment styles come from different attachment experiences of individuals and parents. To promote secure attachment styles, parents need to reassure, comfort, and protect their children in times of stress [65]. PPF refers to the ability of parents to accept, understand and respect their children fully and to timely discover and correct inappropriate parenting attitudes and behaviors [13]. PPF is a protective factor for a healthy parent–child relationship. It helps children improve their emotion regulation to avoid the influence of external negative stimuli [43], thus reducing the risk of externalizing and internalizing behavior problems. Preschoolers are in the initial stage of physical and mental development and are easily affected by external stimuli [66]. Therefore, preventing preschoolers’ behavior problems is crucial, especially among the vulnerable groups of rural preschoolers. Scholars tried to prevent CBP in the past by improving parenting practices and parent–child relationships. However, we ignore the role of the safeguard factor of good parenting practices, namely parenting psychological flexibility. Children’s self-regulation of external stimuli needs to receive more attention from us. Our findings provide new perspectives and empirical support for improving the behavior problems of rural preschoolers.

### 4.3. Limitations and Future Directions

There are still some areas to be improved upon in this study. First, we should pay attention to the parental regulation of psychological flexibility. Due to the limitations of energy and duration in this study, we did not explore this issue. Follow-up studies could explore this in depth. Second, the questionnaire subjects were only caregivers of preschool children in rural areas of three provinces in China. Thus, future research should focus on expanding the sample size, especially for individuals from different cultural contexts. Third, this study obtained all relevant information through questionnaires completed by caregivers. Moreover, due to the limitation of survey tools and energy, we only investigated the contents of disciplinary violations, aggressive behaviors, social withdrawal, depression, and anxiety among children’s externalizing and internalizing behavior problems. However, future research should use multiple methods and sources to obtain relevant information. Finally, this study was cross-sectional and lacked a longitudinal study in the research design. Therefore, longitudinal studies should be added to future studies to improve the reliability of the conclusions.

## 5. Conclusions

Despite the above limitations of our study, to the best of our knowledge, our study is the first to explore how PPF and CER affect CBP, especially for rural preschoolers. Our findings confirm that PPF and two dimensions of CER (emotional regulation and stability) negatively predict children’s externalizing and internalizing behavior problems. Moreover, emotional regulation and stability mediate the relationship between PPF and children’s externalizing and internalizing behavior problems.

Our findings provide a new perspective for improving CBP and bring some implications to family education in rural China and even underdeveloped areas. We encourage the provision of necessary educational and social support for parents and children to enhance PPF and children’s emotional regulation. Furthermore, we also encourage parents to focus on and exercise psychological flexibility. In conclusion, our findings provide a new approach to improving and preventing CBP.

## Figures and Tables

**Figure 1 ijerph-19-15788-f001:**
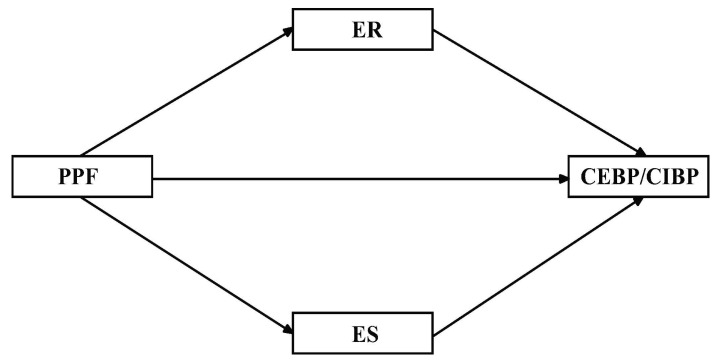
Hypothetical model diagram of the effects of parental psychological flexibility on children’s externalizing and internalizing behavior problems. PPF = parental psychological flexibility, ER = emotional regulation, ES = emotional stability, CEBP = children’s externalizing behavior problems. CIBP = children’s internalizing behavior problems.

**Figure 2 ijerph-19-15788-f002:**
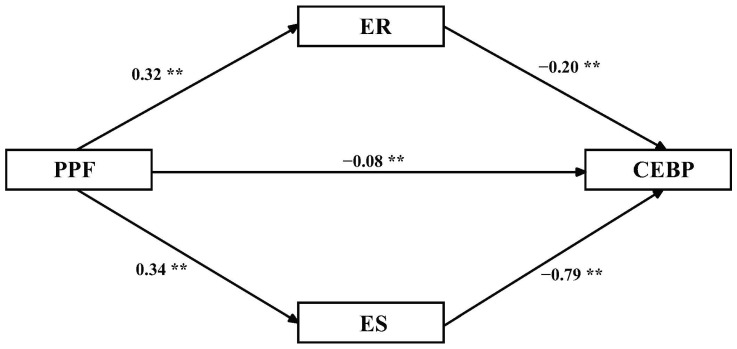
Model diagram of the effect of parental psychological flexibility on externalizing behavior problems. ** *p* < 0.01; PPF = parental psychological flexibility, ER = emotional regulation, ES = emotional stability, CEBP = children’s externalizing behavior problems.

**Figure 3 ijerph-19-15788-f003:**
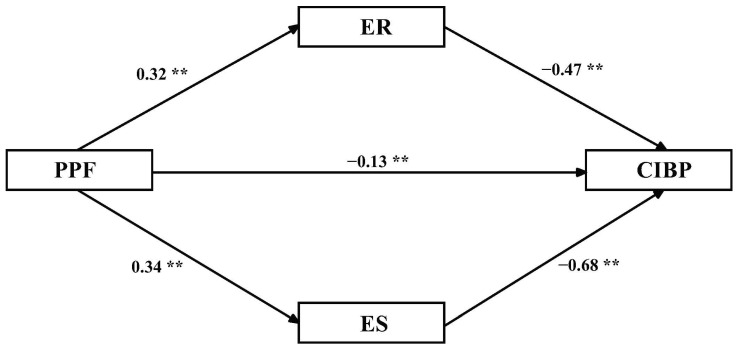
Model diagram of the effect of parental psychological flexibility on internalizing behavior problems. ** *p* < 0.01; PPF = parental psychological flexibility, ER = emotional regulation, ES = emotional stability, CIBP = children’s internalizing behavior problems.

**Table 1 ijerph-19-15788-t001:** Descriptive statistics and association analysis among variables.

Variables	M ± SD	1	2	3	4	5
1 PPF	3.48 ± 0.62	1	--	--	--	--
2 ER	2.39 ± 0.41	0.50 **	1	--	--	--
3 ES	1.91 ± 0.45	0.49 **	0.72 **	1	--	--
4 CEBP	0.97 ± 0.52	−0.41 **	−0.64 **	−0.78 **	1	--
5 CIBP	0.94 ± 0.59	−0.54 **	−0.76 **	−0.81 **	0.85 **	1

Note: ** *p* < 0.01; PPF = parental psychological flexibility, ER = emotional regulation, ES = emotional stability, CEBP = children’s externalizing behavior problems, CIBP = children’s internalizing behavior problems.

**Table 2 ijerph-19-15788-t002:** Regression analysis of externalizing behavior problems.

Regression Equation		Overall Fit Index			Significance of Regression Coefficients	
Result Variables	Predictive Variables	R	R^2^	F	β	t
CEBP	CC	0.81	0.65	73.71	−0.05	−1.22
	CAG				0.04	0.85
	CE				0.16	5.65 **
	CJ				−0.02	−0.62
	CA				−0.01	−0.39
	CG				−0.04	−1.16
	PPF				−0.08	−2.32 **
	ER				−0.20	−3.41 **
	ES				−0.79	−14.55 **

Note: ** *p* < 0.01; CC = caregiver category, CAG = caregiver gender, CE = caregiver education, CJ = caregiver job, CA = child age, CG = child gender, PPF = parental psychological flexibility, ER = emotional regulation, ES = emotional stability, CEBP = children’s externalizing behavior problems.

**Table 3 ijerph-19-15788-t003:** Regression analysis of internalizing behavior problems.

Regression Equation		Overall Fit Index			Significance of Regression Coefficients	
Result Variables	Predictive Variables	R	R^2^	F	β	t
CIBP	CC	0.85	0.72	103.86	−0.01	−0.23
	CAG				0.02	0.41
	CE				0.01	0.23
	CJ				−0.03	−0.89
	CA				0.01	0.22
	CG				−0.02	−0.58
	PPF				−0.13	−3.56 **
	ER				−0.47	−7.90 **
	ES				−0.68	−12.43 **

Note: ** *p* < 0.01; CC = caregiver category, CAG = caregiver gender, CE = caregiver education, CJ = caregiver job, CA = child age, CG = child gender, PPF = parental psychological flexibility, ER = emotional regulation, ES = emotional stability, CIBP = children’s internalizing behavior problems.

**Table 4 ijerph-19-15788-t004:** Mediation analysis of externalizing behavior problems.

Path	Mediating Effect Value	Directing Effect Value	Total Value	Percentile 95% CI	
				Lower	Upper
Path1 PPF ≥ ER ≥ CEBP	−0.06	−0.08	−0.41	−0.12	−0.03
Path2 PPF ≥ ES ≥ CEBP	−0.27	−0.08	−0.41	−0.40	−0.24

Note: PPF = parental psychological flexibility, ER = emotional regulation, ES = emotional stability, CEBP = children’s externalizing behavior problems.

**Table 5 ijerph-19-15788-t005:** Mediation analysis of internalizing behavior problems.

Path	Mediating Effect Value	Directing Effect Value	Total Value	Percentile 95% CI	
				Lower	Upper
Path1 PPF ≥ ER ≥ CIBP	−0.15	−0.13	−0.50	−0.20	−0.11
Path2 PPF ≥ ES ≥ CIBP	−0.23	−0.13	−0.50	−0.31	−0.18

Note: PPF = parental psychological flexibility, ER = emotional regulation, ES = emotional stability, CIBP = children’s internalizing behavior problems.

## Data Availability

The data that support the findings of this study are contained within the article and available from the corresponding author upon reasonable request.

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
