# Peer review of "Parental Psychological Flexibility and Children’s Behavior Problems in Rural Areas in Northeast China: The Mediation of Children’s Emotion Regulation"

_ijerph, 2022, doi:10.3390/ijerph192315788_

Round 1
Reviewer 1 Report
Thank you for the opportunity to review this manuscript.
The paper deals with a timely topic, that is the effects of parental psychological flexibility and children’s emotion regulation on children’s behavior problems. Specifically, the paper investigates the significant role of emotional regulation in this process, using mediation analysis models.
The authors present a well-designed and well-written paper. I really appreciated the use of figures, that clearly visually show the relation between independent and dependent variables, as well as the mediation model.
Methodology is adequate and clearly explained. What remains little unclear is how participants were informed about the research. Who contacted the parents of school attendants?
Discussion section is complete as well, including also limitation section.
What is not very clear about the paper is the novelty of the paper and what is it contribution to the existing literature. This aspect is relevant, because it affects scientific soundness of the paper and its impact on literature of the field.
To this aim, the authors could ameliorate the Introduction section, in order to give more strength to the paper, better underlining and specifying in what this study is new, clarifying in what it contributes to the existing literature. Consistently they could better underline, in the Discussion and Conclusion section, the novelty of results from the study, therefore indicating its strengths.
Finally, two simple remarks:
Line 28-29: please provide a reference for this
Line 46: please specify what CER stands for
After these minor revisions, in my opinion the manuscript is suitable for publication
Author Response
Response to Reviewer 1 Comments
Dear Reviewer:
Thank you for the reviewer’s comments concerning our manuscript entitled “Parental Psychological Flexibility and Children's Behavior Problems in Rural Areas in Northeast China: The Mediation of Children's Emotion Regulation” (ID: ijerph-2037929). They are all valuable and constructive comments for revising and improving our paper and guiding our research. We have corrected according to every comment. All of the modified portions responding to the reviewer’s comments are marked in yellow. The corrections in the paper and the responses to the reviewer’s comments are as following:
Point 1: What remains little unclear is how participants were informed about the research. Who contacted the parents of school attendants?
Response 1: A brief description was added in our participants (Lines 190-193) and procedure (Lines 244-250).
Point 2: What is not very clear about the paper is the novelty of the paper and what is it contribution to the existing literature.
Response 2: In the third paragraph (Lines 79-81) and the last paragraph (Lines 172-174) of the introduction, the fifth paragraph of the discussion (Lines 419-420), and the second paragraph of the conclusion (Lines 443-448), we try to add novelty and contributions of this study.
Point 3: two simple remarks(1)Line 28-29: please provide a reference for this (2)Line 46: please specify what CER stands for
Response 3: We added remarks in the introduction’s first paragraph (Lines 35-38) and the sixth paragraph (Lines 119-122).

Reviewer 2 Report
This is a very interesting research, but the presentation in this paper, is very difficult to read and is not really respecting usual writing for scientific papers.
The excessive use of acronymes makes reading very unpleasant. The authors must find a solution to help the reader and force him to go back to definitions 'that deserve to presented in a table).
There is a continuous mixture of material, methods and results, into the Mat and Meths section, and there are Discussions into Results. As an example, all data describing the enrolled population (sex, age, ...) must be in the results section. The authors must edit it and put each information in its section. The section about Statistics, that is not named this way, must be re-written. We discover information about the statistics, all along the Results section, this is not the way to present it and it creates some confusion. The paper is so difficult to read, that I still have some doubts about the accuracy of some statistical analyses.
The Discussion section is not really a discussion, since the authors are just pushing their initial hypothesis and do not really challenge it. A more structured Discussion would provide more value to the paper.
The Abstract needs to be re-written too. There is a dramatic lack of data in it and not enough information about the methods used in this study and the results (measured ones).
Regarding the Discussion and the Introduction, it is quite strange that a research about factors influencing the adaptability (including the emotional balance) of young children (pre-scholar), do not say a word about the attachment and its quality (e.g. "Impact of attachment, temperament and parenting, on human development. Yoo Rha Hong & Jae Sun Park; 2012 ,or "The association between attachment patterns and parenting styles with emotion regulation among Palestinian preschoolers." Qutaiba Agbaria et al, 2021). The authors should, at least, discuss this question and not just pay attention to the parents' psychological flexibility. The writing of the paper, as well as the theoretical approach described into the Introduction and Discussion-Conclusion sections, give the impression of an excessive simplification of the question. Moreover, the authors do not justify their decision to use the PPF measurement for a study with young children (3-8 years-old). The research and validation for the use of this concept is more related to families with pre-adolescent and adolescent children (10-18 years-old) (Burke and Moore, 2015). I also have some problems with CBP. It is quite a broad category, including so different conditions (anxiety-related disorders, opponent attitudes, sleep disorders,...). Using such a vague category to calculate correlations with other parameters, could be considered as a risk to find correlations, whatever the study and population studied. The authors should, at least, clarify and justify this strategy.
Author Response
Response to Reviewer 2 Comments
Dear Reviewer:
Thank you for the reviewer’s comments concerning our manuscript entitled “Parental Psychological Flexibility and Children's Behavior Problems in Rural Areas in Northeast China: The Mediation of Children's Emotion Regulation” (ID: ijerph-2037929). They are all valuable and constructive comments for revising and improving our paper and guiding our research. We have corrected according to every comment. All of the modified portions responding to the reviewer’s comments are marked in yellow. The corrections in the paper and the responses to the reviewer’s comments are as following:
Point 1: The excessive use of acronymes makes reading very unpleasant. The authors must find a solution to help the reader and force him to go back to definitions 'that deserve to presented in a table).
Response 1: We are very sorry for the inconvenience caused by the excessive use of acronymes. We reduced the acronymes to the following three: PPF = parental psychological flexibility, CER = children's emotion regulation, and CBP = children's behavior problems. And below each table and figure, we add notes (Lines 183-186; Lines 282-284; Lines 301-303; Lines 305-307; Lines 320-321; Lines 322-324; Lines 336-337; Lines 338-340). Then, if the paper is still reading unpleasant, we can stop using acronymes.
Point 2: There is a continuous mixture of material, methods and results, into the Mat and Meths section, and there are Discussions into Results. As an example, all data describing the enrolled population (sex, age, ...) must be in the results section. The authors must edit it and put each information in its section. The section about Statistics, that is not named this way, must be re-written. We discover information about the statistics, all along the Results section, this is not the way to present it and it creates some confusion. The paper is so difficult to read, that I still have some doubts about the accuracy of some statistical analyses.
Response 2: We rewrote the materials and methods part and divided it into participants, measures, procedures, and data analysis. We included the population information in the participants to illustrate the basic situation of the survey sample (Articles such as “Mediating effect of sequential memory on the relationship between visual-motor integration and self-care performance in young children with autism spectrum disorder” and “Parents’ Perceived Social Support and Children’s Approaches to Learning in Rural China: A Moderated Mediation Model” also put population information into the participants). We changed the title and deleted the statistical description in the results (Lines 271-273; Lines 285-287; Lines 308-310). Finally, we rewrote the data analysis (Lines 255-263; Lines 266-269), presenting the statistical analysis methods used in this study.
Point 3: The Discussion section is not really a discussion, since the authors are just pushing their initial hypothesis and do not really challenge it. A more structured Discussion would provide more value to the paper.
Response 3: We rewrote the discussion. First, we analyze the differences between our findings and existing studies on the effect of parental psychological flexibility on children's behavior problems (Lines 354-365). Second, based on the existing research, we expand our interpretation of emotional regulation and stability affect children's behavior problems, especially for rural preschoolers (Lines 375-386). Third, to discuss the mediating effect of emotion regulation, we tried to explain our findings in terms of the ecosystem theory and the theory of cognitive development (Lines 388-401). Forth, we pay attention to the influence of family education, parent-child relationship, and individual initiative on individual development, especially for rural preschoolers (Lines 402-417). Moreover, we also put forward the future research direction (Lines 423-435).
Point 4: The Abstract needs to be re-written too. There is a dramatic lack of data in it and not enough information about the methods used in this study and the results (measured ones).
Response4: We rewrote the abstract. We added data, methods, and results (Lines 11-19; Lines 23-24).
Point 5: Regarding the Discussion and the Introduction, it is quite strange that a research about factors influencing the adaptability (including the emotional balance) of young children (pre-scholar), do not say a word about the attachment and its quality (e.g. "Impact of attachment, temperament and parenting, on human development. Yoo Rha Hong & Jae Sun Park; 2012 ,or "The association between attachment patterns and parenting styles with emotion regulation among Palestinian preschoolers." Qutaiba Agbaria et al, 2021). The authors should, at least, discuss this question and not just pay attention to the parents' psychological flexibility. The writing of the paper, as well as the theoretical approach described into the Introduction and Discussion-Conclusion sections, give the impression of an excessive simplification of the question.
Response 5: We rewrote the introduction and discussion to analyze the research questions in depth. In the individual influencing factors, we added the content of the influence of attachment on children's behavior problems (Lines 69-74). We try to explain the value of emotional regulation as a mediator (Lines 74-81). Moreover, In the discussion, we add attachment studies to provide explanations and support for our findings (Lines 403-408).
Point 6: Moreover, the authors do not justify their decision to use the PPF measurement for a study with young children (3-8 years-old). The research and validation for the use of this concept is more related to families with pre-adolescent and adolescent children (10-18 years-old) (Burke and Moore, 2015). I also have some problems with CBP. It is quite a broad category, including so different conditions (anxiety-related disorders, opponent attitudes, sleep disorders,...). Using such a vague category to calculate correlations with other parameters, could be considered as a risk to find correlations, whatever the study and population studied. The authors should, at least, clarify and justify this strategy.
Response 6: We added evidence to the measures to analyze the possibility of assessing preschoolers’ parental psychological flexibility (Lines 211-215).
We revised the statements in the manuscript, and we revised some complicated sentences in the manuscript. In addition, we proofread the spelling, grammar, diction, and other problems in the manuscript through “AJE” .

Round 2
Reviewer 2 Report
The paper has been considerably improved and is now easier to read. Thank you.